# Metamaterial bricks and quantization of meta-surfaces

Gianluca Memoli[1], Mihai Caleap[2], Michihiro Asakawa[1], Deepak R. Sahoo[1], Bruce W. Drinkwater[2] & Sriram Subramanian[1]

Controlling acoustic fields is crucial in diverse applications such as loudspeaker design, ultrasound imaging and therapy or acoustic particle manipulation. The current approaches use fixed lenses or expensive phased arrays. Here, using a process of analogue-to-digital conversion and wavelet decomposition, we develop the notion of quantal meta-surfaces. The quanta here are small, pre-manufactured three-dimensional units—which we call metamaterial bricks—each encoding a specific phase delay. These bricks can be assembled into meta-surfaces to generate any diffraction-limited acoustic field. We apply this methodology to show experimental examples of acoustic focusing, steering and, after stacking single meta-surfaces into layers, the more complex field of an acoustic tractor beam. We demonstrate experimentally single-sided air-borne acoustic levitation using meta-layers at various bit-rates: from a 4-bit uniform to 3-bit non-uniform quantization in phase. This powerful methodology dramatically simplifies the design of acoustic devices and provides a key-step towards realizing spatial sound modulators.

[1] INTERACT Lab, School of Engineering and Informatics, University of Sussex, Brighton BN1 9RH, UK. [2] Department of Mechanical Engineering, University of Bristol, Bristol BS8 1TR, UK. Correspondence and requests for materials should be addressed to G.M. (email: g.memoli@sussex.ac.uk).

The shaping of light using spatial light modulators (SLMs) is an established technology for advanced three-dimensional (3D) displays[1] and micro-manipulation[2]. In the SLM an incident beam of coherent light is transformed via amplitude and phase manipulation into a wide range of reflected or transmitted optical distributions. Crucially the SLM is computer controlled and it is possible to reconfigure the optical field produced in almost real-time.

The acoustic equivalent of an SLM does not exist. Traditionally, the control of an acoustic field distribution was achieved by using fixed lenses[3], which perform a single function, or phased arrays[4,5], where the amplitudes and phases of the individual array elements are independently controlled. However, phased arrays are often bulky and expensive, with cost and complexity scaling linearly with the number of channels. Despite these limitations, phased arrays are in widespread use. In High Frequency Focused Ultrasound (HIFU), for example, sparse arrays of transducers are used to treat a variety of tumours[6] or functional brain disorders[7], inducing a localized heating effect, even behind the ribs[8]. In industrial applications, focusing and steering of ultrasonic waves is required to find small cracks in metallic components, which can be complex in geometry and highly anisotropic[9]. New applications that require precise control of acoustic waves include parametric loudspeakers[10], ultra-haptics[11], caustic engineering and acoustic levitation[12].

Acoustic metamaterials are an emerging class of engineered materials designed to control, direct and manipulate acoustic waves[13]. Typically, they are made up of a collection of sub-wavelength structures (that is, unit cells), and are characterized by their effective mass density and bulk modulus. The possibility of manufacturing metamaterials with negative effective parameters[14] has led to effects such as negative refraction[15] and sub-diffraction focusing. Particularly interesting for beam-shaping applications are two-dimensional (2D) planar meta-surfaces: closely-packed structures of phase shifters whose thickness is comparable to the wavelength of operation. Very recent examples of acoustic meta-surfaces include the use of labyrinthine structures[16], helical structures[17], space-coiling[18,19], multi-slits[20,21] and Helmholtz resonators[22–25]. These diverse meta-surfaces have always been built using a limited number of unit cells, and as such the optimal number of phase discretization levels required for each application has been ignored. For instance, previous studies[16–29] suggest a wide range of discretization levels, varying from 2 to 256, selected on an *ad-hoc* basis.

Here we develop the notion of quantal meta-surfaces to demonstrate a different metamaterial concept, based on the use of a small set of pre-manufactured 3D unit cells, termed metamaterial bricks, which can be assembled into 2D structures on-demand. The bricks become, in isolation, the building blocks of an assembly, encoding prerequisite phase delays. This operation is a form of analogue-to-digital conversion (Fig. 1): the desired acoustic pressure field is sampled at a certain distance from a meta-surface and used as input for acoustic holography, leading to a phase distribution that gets quantized in the spatial and phase domains, whose values are then mapped into a series of pre-manufactured metamaterial bricks. Starting from a limited set of unique bricks, we use a discrete wavelet transform based method to synthesize the meta-surface needed in a given application, optimizing the number of bricks needed. We obtain a reconfigurable meta-surface, which transforms an incident sound wave into an arbitrary range of diffraction-limited acoustic fields. Furthermore, once a number of meta-surfaces are available, each performing a given transformation (for example, steering and focusing), we show that single meta-surfaces can be stacked into layers to perform more complex transformations, thus creating the acoustic equivalent of optical components. This simple, yet powerful, concept simplifies the design of

acoustic devices and systems, and lays the foundations for realization of spatial sound modulators (SSMs).

## Results

**Meta-surface quantization.** A number of studies have explored how to encode a uniform phase distribution $\varphi(x, y)$ in a meta-surface to produce given acoustic transformations[13,14]. Realizing a quantal meta-surface, however, requires AD conversion with two parameters: one in the spatial domain, which—for a fixed size of meta-surface—depends on the size of its unit cells and on how the phase $\varphi_{ij}$ is assigned to each location ($x_i$, $y_j$), and one in the phase domain, which governs the number of different phases. Practically, the AD step defines the parts list and assembly instructions for a particular meta-surface. We chose a spatial resolution of $\lambda_0/2$, which is a good compromise between the ease-of-manufacture and the need to realize diffraction-limited fields without spatial aliasing[4]; here, $\lambda_0$ denotes the operating wavelength.

Different uniform mappings of the phase domain have been attempted in the past, with acoustic examples raging between 2- and 3-bit[16,18,19] and electromagnetic studies down to 1-bit[26,27,29]. None of these works, however, have discussed how the choice of the quantization level impacts on the fidelity with which the desired field is reproduced (Supplementary Note 1). Here we treat the phase distribution $\varphi_{ij}$ like a 2D image, so that optimization of the AD conversion stage—that is, the process that, given a desired precision in the acoustic field, minimizes the number of different phases to be used and, possibly, the number of elements for each phase (while maintaining the given precision in the reproduced field)—is then analogous to a vector quantization in image compression. Different compression methods are possible (everyone being familiar with classical JPEG protocol), but wavelet-based methods[30] are specifically aimed at determining the lowest number of coefficients necessary for a specified reconstruction quality of localized features[31,32], like the abrupt changes of phase in a focusing meta-surface (Supplementary Note 1). A discrete wavelet transform (DWT) represents the image over different scales, selecting at each step the key features (low spatial frequencies) and the residual ones (high-spatial frequencies). This generates a hierarchical tree of matrices, where the spatial resolution doubles at each step (Supplementary Fig. 1). Once the tree is obtained, the compression procedure works as follows: $\varphi_k^{DWT}$ is computed, up to level $K$, and the coefficients below a certain threshold value $\delta$ are set to zero. The inverse transform $\tilde{\varphi}_{ij}$ is finally computed, containing in general less information and a smaller number of required phases, which are crucially not uniformly distributed either spatially or in the phase domain.

Figure 2a,b shows the phase distribution to form a focus at (0,0,100) mm (see Methods), based on a $16 \times 16$ grid, where we calculated the DWT using the classic Haar function as parent wavelet (Supplementary Fig. 1). Note here each pixel in Fig. 2a, that is, the original uncompressed image, is $\lambda_0/2$ in size. As shown in Fig. 2c, the number of unique phases, that is, quanta, needed to realize $\tilde{\varphi}_{ij}$, decrease with increasing $\delta$: 8 quanta (3-bit) are sufficient for a compression rate of 4:1, 6 quanta at 4.6:1 and 4 quanta (2-bit) at 8:1. As expected, for all cases, the quanta values are non-uniformly distributed in the interval $[0, 2\pi]$. To select the thresholding levels to use in the following experiments, we compute the error in approximating the continuous phase distribution with the sum of the squared differences:

$$\varphi_E(\delta) = \frac{1}{N} \sqrt{\sum_{ij} \left[\varphi_{ij} - \tilde{\varphi}_{ij}(\delta)\right]^2} \qquad (1)$$

Since this quantity is directly related with the precision with which the desired field can be realized (Supplementary Note 2),

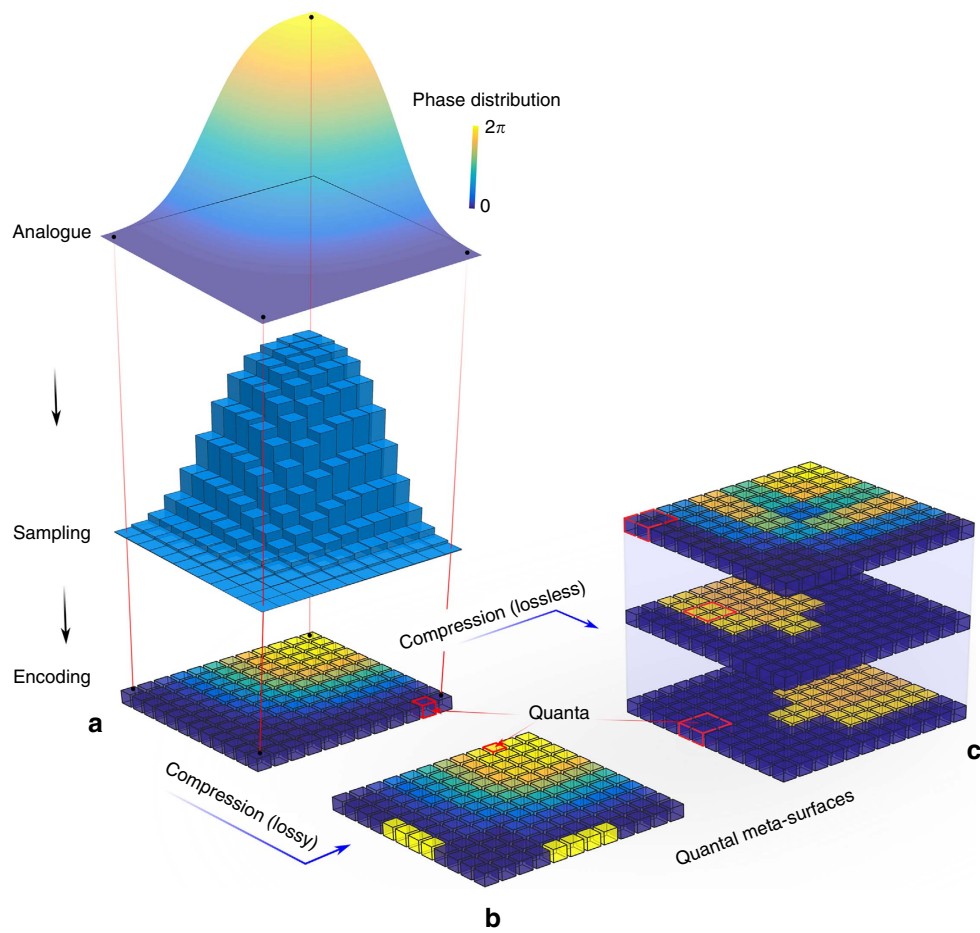

**Figure 1 | Notion of quantal meta-surface inspired from analogue-to-digital conversion and image compression.** (**a**) Quantization of an analogue phase distribution with a uniform $2\pi$-span and a fixed spatial resolution. (**b**) Lossy and (**c**) lossless compressions of figure (**a**) using wavelet transforms, with and without thresholding, respectively. In this example, the lossless alternative results (at first level) in a three branch tree to represent the structure of the decomposition; note here the spatial resolution doubles. Both, (**b,c**) contain less information and a smaller number of quanta—which are no longer uniformly distributed either spatially or in the phase domain.

we target the value $\varphi_E = 0.25$ radians, which corresponds to the error in the phase distribution obtained using a uniform 4-bit quantization, leading to a 0.1 dB error on the pressure distribution of a focusing meta-surface (Supplementary Note 2; Supplementary Fig. 2). As shown in Fig. 2d, both $\delta = 1/8$ (11 quanta) and $\delta = 3/16$ (8 quanta) fulfil this condition and will be realized experimentally in the following sections. Errors of 1 dB are expected with 8 uniformly distributed quanta (3 dB with 4 uniformly distributed quanta), and they may become as large as 8 dB when 1-bit solutions are used (Supplementary Fig. 2).

To properly assign the desired quantized phase values resulting from this analysis, we sculpt next the geometry of our quanta, which we call metamaterial bricks. As we will show in the rest of the presented work, these metamaterial bricks can be used to build various different structures and acoustic devices, emphasizing the power of the simplicity of this concept.

**Metamaterial bricks**. To form a desired acoustic field with exceptional performance, the metamaterial bricks should possess the ability to transmit sound effectively, locally shift phase with a $2\pi$ range, and hold sub-wavelength spatial resolution to avoid spatial aliasing effects[33]. By spatially tailoring the geometry of the bricks in a planar meta-surface, one can modulate the transmitted waves in nearly arbitrary ways in a specific frequency range.

While most of the other studies explore the audible range, our metamaterial bricks are designed for operation in the ultrasonic range at 40 kHz (wavelength $\lambda_0 \approx 8.66$ mm in air at 25 °C). As shown in Fig. 3a, each brick appears geometrically like a rectangular cuboid with a square base-shape of side $\lambda_0/2$ and a height of $\lambda_0$, and consists of an open central channel that delays the incident wave, hence shifting the relative phase of the output. The channel topology is designed to be suitable for microfabrication (see Methods) and was inspired by a pool of alternative designs[16–25], where some of these are based on the pioneering work on coiling space by Liang and Li[34]. Fully 3D bricks are constructed by extruding the surface of four parallel bars (of variable spacing and length) positioned orthogonally to the wave direction, thereby creating a labyrinth meander. The inter-bar spacing $b_s$ and bar length $b_l$ can be effectively tuned, resulting in a phase shift covering a $2\pi$ span (Supplementary Fig. 3). Simulations showed that a uniform 4-bit quantization of the phase space can reproduce any focused field (with a focal length between $3\lambda_0$ and $47\lambda_0$) with an error of $< 0.1$ dB (Supplementary Fig. 2), so sixteen bricks were designed—corresponding to delays of $0, \pi/8, \ldots, 15\pi/8$ in phase. Figure 3b shows full-wave simulations (see Methods) of the pressure field distributions when a plane wave travels through each of the 16 selected bricks. While the design of the bricks is not unique (Supplementary Note 3), an important feature of the

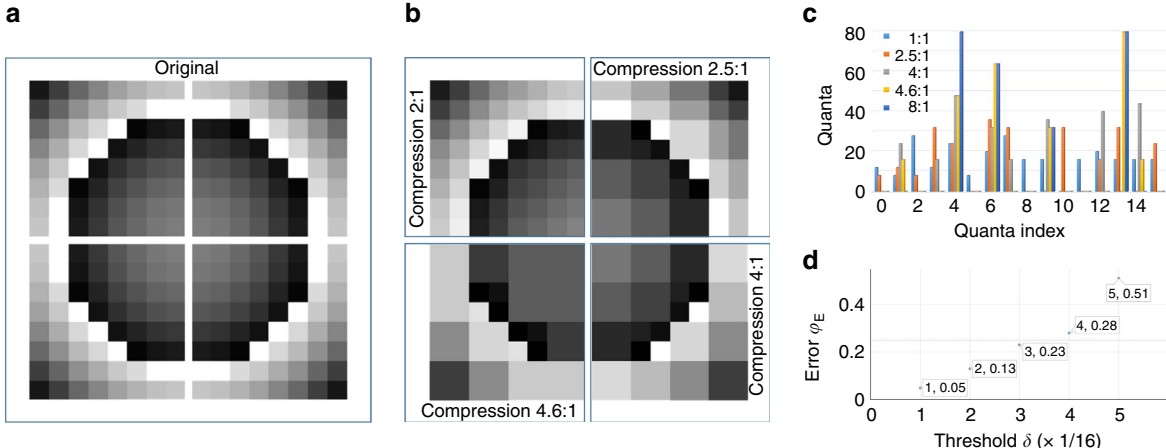

**Figure 2 | Example of image compression with various thresholding levels.** (**a**) Original image with $16 \times 16$ resolution; here, the number of phase values (that is, quanta) is sixteen. (**b**) Compressed image for different values of the threshold $\delta$ Only quarter-images are shown (due to symmetry) compressed respectively at $2{:}1 - \delta = 1/16$, $2.5{:}1 - \delta = 1/8$, $4{:}1 - \delta = 1/4$ and $4.6{:}1 - \delta = 1/2$. The corresponding values of the phase error $\varphi_E(\delta)$ are reported in figure (**d**) while the distribution of unique quanta versus quanta index is shown in **c**.

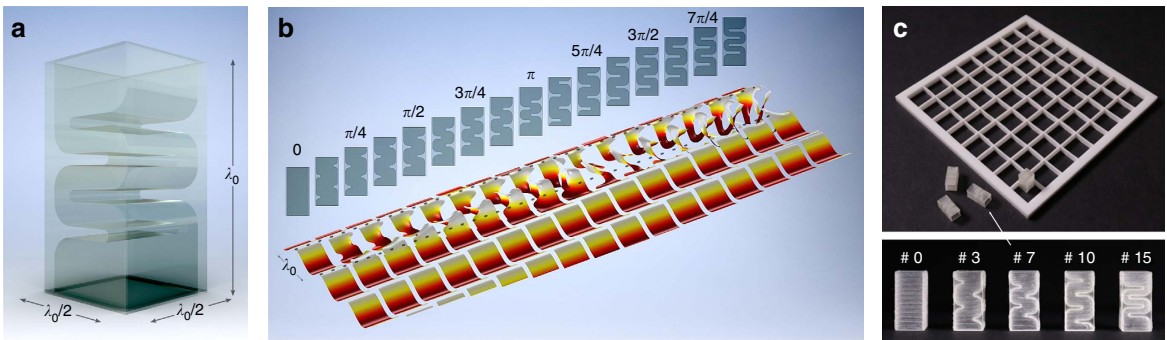

**Figure 3 | Metamaterial bricks.** (**a**) 3D rendering of a brick. (**b**) Cross-sections of 16 selected bricks and the corresponding phase maps at normal incidence. Each case is calculated independently by impinging a plane wave with a wavelength $\lambda_O$ through the bricks (located in between the two dashed lines), clearly showing a $2\pi$ span of the transmitted phase. Geometrical parameters for each brick are shown in Supplementary Table 1. (**c**) Photograph of the fabricated bricks and the grid to contain them. The numbers at the top of each brick denote the corresponding phase shift (in units of $\pi/8$). See also Supplementary Video 1.

selected geometry is that the effective acoustic impedance of each brick is matched to that of air, increasing the efficiency of wave transmission and suppressing reflection. In fact, the average transmission magnitude over all units, according to our full-wave simulations, is $>98\%$ (Supplementary Table 1). We then manufactured in advance, by rapid prototyping, a set of 16 types of bricks, which could be mounted into a laser-cut grid frame structure, where each grid-square contains 4 bricks in a $2 \times 2$ assembly (Fig. 3c; Supplementary Video 1). Transmission measurements for three selected bricks (see Methods) confirmed an experimental average transmission of $97 \pm 5\%$, within 10% from the predicted values. Note that functionally, each brick presents a directivity pattern, which depends on its internal topology and on the frequency used. Hence, to reduce unwanted steering effects, the bricks were assembled with a random orientation.

Through a series of numerical simulations and accompanying experiments, we now demonstrate the utility of the concept of quantal meta-surfaces in the design of several structures with special functionalities. Specifically, we synthesize meta-surfaces, which in isolation apply simple transformations (for example, beam steering, focusing) to a plane incident beam. The low-transmission loss means that quantal meta-surfaces can also be stacked into layers to perform additive transformations; for

example, a focusing layer can be combined with a beam-steering layer, to form an off-axis focus, or with an inverting annulus, thus creating a bottle-shaped field. The latter configuration is used to successfully levitate a small polystyrene bead at different bit-rates (from 4-bit to 3-bit quantization). While any diffraction-limited field can be theoretically created with just a single layer, stacking brings about additional conveniences for the concept of spatial sound modulators. Due to the additivity of phase delays, each meta-surface in the stack can in fact be realized with a lower bit-rate, so that a smaller number of brick types is eventually needed. This advantage is particularly evident when the phase quantization is non-uniform.

**Steered and focused beams.** Using the metamaterial bricks shown in Fig. 3, we now design a blazed transmission grating. Previous studies on reflection/refraction/beam steering with meta-surfaces[16,19,21,25] were typically performed using a saw-tooth phase gradient along the output interface. The observed effects were explained in terms of a generalized version of the Snell's law[27], valid only for slow variations of the phase $\varphi(x)$. However, spatial discretization of a meta-surface means that the phase variation is a series of steps, and so the behaviour is more completely explained in terms of diffraction theory[35].

The space discretized phase-ramp is an acoustic diffraction grating that produces beam steering at an angle dependent on the wavelength and on the spatial period, $d$, over which the range $[0, 2\pi)$ is covered, that is, a 'line'[36]. In this configuration, the energy is split between the desired order (typically $m = 1$) and the other orders of diffraction (mainly $m = 0$ and $m = -1$), which correspond to unwanted directions of propagation. It is worth noting that, when the energy goes mostly into $m = -1$, one gets what is called negative refraction; in many studies, this is certainly a desired effect. Tuning the gradient of the phase over the line width $d$, a very efficient energy transfer to the first order can be achieved under certain conditions (that is, blazing), so that in practice only one beam is present, as described by generalized Snell's law. In optics, blazed gratings are similarly realized with a nematic SLM: the steering angle is determined by the spacing between the lines $d$, but the gradient of the phase in each line can be optimized to maximize the energy in a single output beam.

Here, we demonstrate the strength of the diffraction-based approach by realizing an experiment that cannot be explained by the generalized Snell's law. Each line contains a linear phase-ramp and a constant phase section of variable length, covering the full $2\pi$-range. We use two $\lambda_0/2$-bricks (respectively of phases $\varphi_1 = 5/8\pi$ and $\varphi_2 = 11/8\pi$) and a channel of variable width ($\varphi_3 = 2\pi$). In this way the gradient over the first two bricks stays fixed at $\partial\varphi/\partial x \approx 0.69 \cdot 2\pi/\lambda_0$, while the line width $d$ varies between $\lambda_0$ and $2\lambda_0$. Results in Fig. 4a–c confirm that the diffraction angle decreases as line spacing is increased, but also show that energy gets progressively shifted into secondary lobes as this happens. The far-field diffraction pattern of the grating for varying $d$ is compared in Fig. 4d with the theoretical transmitted angles $\theta_t$ calculated with the grating equation, $m\lambda_0 = d \sin \theta_t$, and excellent agreement is found. This is also confirmed by experimental results reported in Fig. 4d, which agree perfectly with both numerical and theoretical results.

Another example with special functionality, to which we apply the notion of quantal meta-surface, is the steering of a focused beam. To achieve an off-axis beam focusing, we combine a focusing meta-surface with a blazed grating in a multi-layered structure. Optical studies[26] have investigated theoretically the concept of multi-layered metamaterials, indicating how stacks of meta-surfaces can be used to create an arbitrary Green's function, to either transform a known input into a desired field and/or to perform mathematical operations (that is, differentiation). Here we build on this concept, providing the experimental realization (in acoustics) thanks to the excellent transmission capabilities of the bricks. Here, we first synthesized a meta-surface with a uniform 4-bit phase quantization for focusing along the axis, at $F_0 = 100$ nm (see Methods). We then mounted a blazed grating

on top of the focusing meta-surface; the inter-layer separation was arbitrarily fixed to $3\lambda_0/4$. The grating meta-surface is a result of the previous analysis, and is formed by three $\lambda_0/2$-bricks, respectively of phases $\varphi_1 = 5/8\pi$, $\varphi_2 = 11/8\pi$ and $\varphi_3 = 2\pi$, whose performance is simulated in Fig. 5a. The measurements of Fig. 5b show that the 4-bit experimental realization performs as predicted by the full-wave simulations. Naturally, we could have used a single meta-surface and 16 brick types to perform the desired operation, but stacking makes it possible to achieve the same result using only 8 unique bricks (identified by the wavelet-based compression algorithm), as shown in Supplementary Fig. 4. Both full-wave simulations and experiments (Supplementary Fig. 4a,b) suggest in fact only small differences in the focusing field obtained with a 4-bit uniform and the 3-bit non-uniform phase quantization identified by wavelet compression; in particular, we note a prolonged focal region with a slightly shifted focus with the low-bit quantal meta-surface. Also, interestingly, the size of the focal region perpendicular to the axis depends on the lateral dimensions of the meta-surface ($8\lambda_0$ in this study): the larger the latter, the tighter the focus. According to the Rayleigh criterion, our source should give a resolution limit of $0.61\lambda_0$, whereas the ½-width of the spot in experiments was found to be $0.85\lambda_0$ (Supplementary Fig. 4c). It is also worth noting that our focus is at 50 mm from the axis and therefore outside the boundaries of the meta-surface, bringing focused energy where other studies[16] did not manage.

**Bottle-shaped beams and acoustic levitation.** The final example deals with the use of quantal meta-surfaces in realizing a bottle trap, which can also be used as a tractor beam. This field can be created by the superposition of a focus and an additional phase screen in which a central circular region is driven out of phase with the outer regions[12]. Such beams have been shown to offer stable single-sided acoustic levitation near the focal region. Figure 6 plots the simulated and measured (see Methods) transmitted field maps corresponding to a stack comprising a focusing quantal meta-surface on top of a phase-inverting annulus. Here, we compare two levels of quantization for the phase distribution: 4-bit and 3-bit. In both cases we observed the expected low-pressure 'quiet' region near the focus ($F_0 = 100$ nm. As in Supplementary Fig. 4, the quiet region was shifted further away from the top meta-surface when the 3-bit phase quantization was applied. These effects were captured both in simulations and measurements, with excellent agreement between the two. For the phase distributions of the synthesized meta-surfaces shown here, Supplementary Fig. 4d. A snapshot demonstrating the acoustic levitation of a polystyrene bead with a diameter in the range of 2.1–2.5 mm is shown in Fig. 6d.

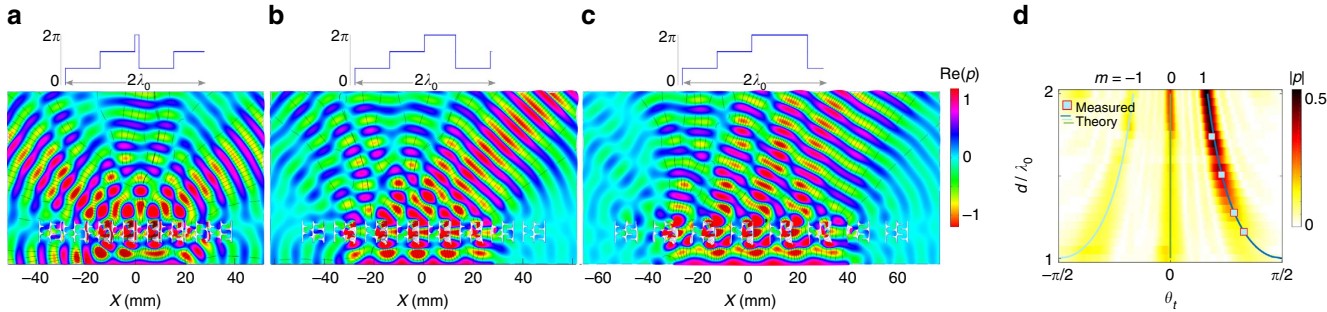

**Figure 4 | Blazed transmission gratings.** (a–c) Transmitted pressure field maps (real-valued) over a blazed grating with variable line widths, $d = 17\lambda_0/15$, $22\lambda_0/15$ and $27\lambda_0/15$, respectively. The sketch above the maps shows the phase profile over $2\lambda_0$. (d) Far-field pressure field map for line widths $d$ varying between $\lambda_0$ and $2\lambda_0$. Theoretical (continuous lines) and experimental (symbols) transmitted angles $\theta_t$ are compared with the numerical simulation. Colour bars represent normalized pressures.

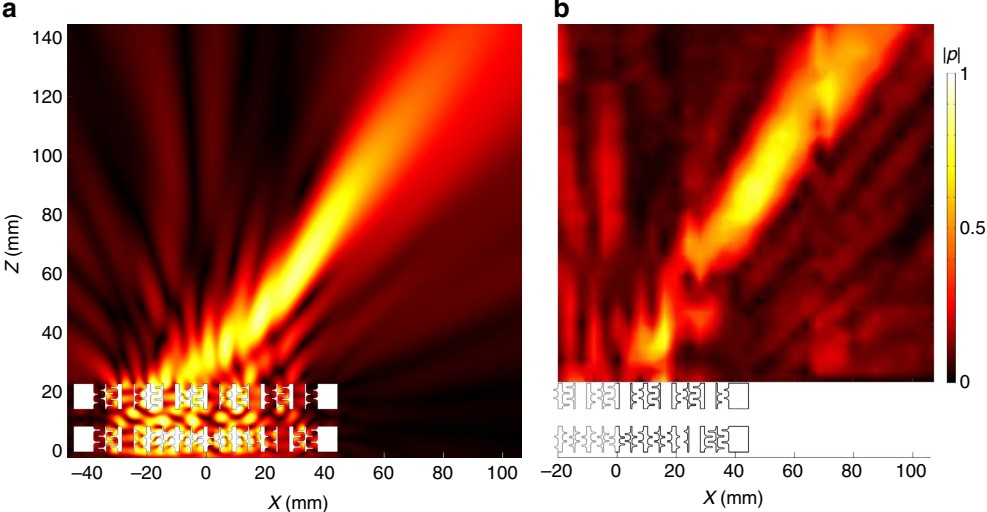

**Figure 5 | Steering of a focused beam with meta-surfaces.** (**a**) Simulation and (**b**) measurement results showing the pressure field maps at $Y = 0$. Colour bars represent normalized pressures.

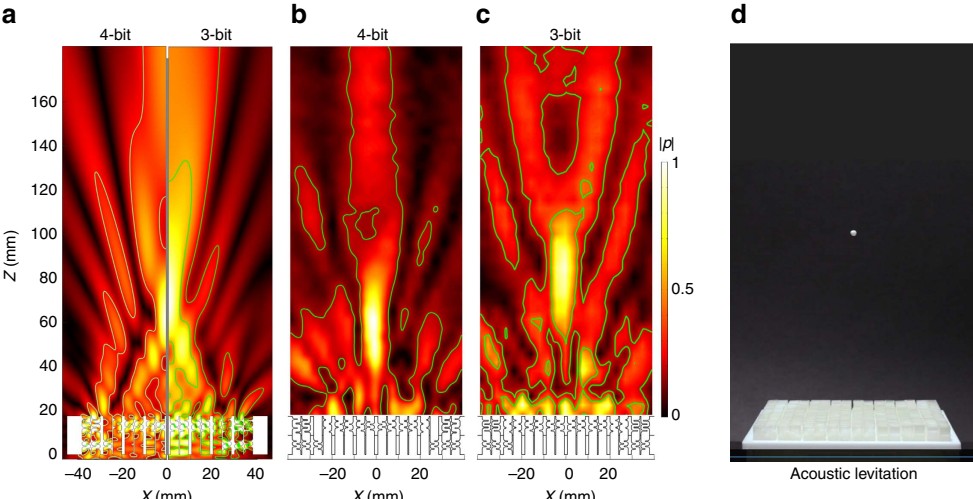

**Figure 6 | Bottle-shaped beams and single-sided acoustic levitation with quantal meta-surfaces.** (**a**) Simulations and (**b,c**) measurements showing the pressure field maps at different bit-rates. Here, the pressure field is probed in the vertical plane $Y = 0$. (**d**) Acoustic levitation of a polystyrene bead using a 4-bit focusing meta-surface. (Snapshot from Supplementary Video 1). Colour bars represent normalized pressures.

The procedure of assembling metamaterial bricks into meta-surfaces, and layers to achieve acoustic levitation of a polystyrene sphere is shown in Supplementary Video 1. Herein, in addition to 4- and 3-bit quantization (16 and 8 unique bricks, respectively), we also show an acoustic levitation demonstration with 11 unique bricks corresponding to a 2.5:1 compression rate of a focusing meta-surface.

## Discussion

The results reported in this paper reveal an algorithm for the fabrication of quantal meta-surfaces with well-controlled arbitrary 2D phase distributions using 3D metamaterial bricks. The concept of our method was inspired from analogue-to-digital conversion and image compression. Here, the AD conversion denotes a process in which a continuously variable (analogue) phase distribution is changed, without altering its essential content, into a multi-level (digital) phase. A quantal meta-surface becomes in effect a digital phase processing system operating as an SLM. As summarized in Fig. 1, three possible routes can be considered to realize a quantal meta-surface. First, we start with a spatial resolution of $\lambda_0/2$ and 4-bit uniform representation of the analogue phase (Fig. 1a). This method is similar to what has been reported by other authors. Second, we treat the phase distribution as an image and use a wavelet transform to perform a lossy compression (Fig. 1b). In this way, the number of bits necessary to represent the analogue phase is reduced to 3, while maintaining the error in reproducing the field below 0.1 dB: this has been explored in Fig. 6 and Supplementary Fig. 4. There is however a lossless alternative (Fig. 1c), where we can exploit the additivity of phase and the hierarchical properties of wavelet transforms. These transforms result in a branched tree to represent the structure of the decomposition; at a given level, each of the branch (that is, meta-surface) can be layered to form a stack with a smaller number of unique bricks. An example where this route is applied to a focusing field is reported in Supplementary Fig. 5. Here, the analogue phase is represented with 3.5 bits and a stack of 4 quantal meta-surfaces. In this solution, each meta-surface is made of 'blocks'—that is, unit cells of dimension $\lambda_0$, each made of a 2 × 2 structure of identical bricks. While the spatial resolution

of the bricks has to remain lower than $\lambda_0/2$ to reduce higher order diffraction, the presence of blocks makes mechanical assembly much simpler, which is a key aspect of cost-effective devices based, for example, on active meta-materials[26–28].

Our metamaterial bricks are designed for operating at 40 kHz, nevertheless the presence of the labyrinth meander means that each brick can also work at lower frequencies[34]. In particular, since the additional phase delay depends on the ratio between the effective length $\ell_{\rm eff}$ of the meander and the wavelength $\lambda_0$ of the incident wave, each selected brick will have the same transmission performance at frequencies $f_j = f_0 - jc_0/\ell_{\rm eff}$; here, $j = 0, 1, 2\ldots$ is an integer, $f_0$ is the design frequency and $c_0$ is the speed of sound. Although not demonstrated herein, full-wave simulations confirm this behaviour. The frequency response, and the potential use of amplitude modulation, would require further studies that go beyond the scope of the present work. If realized, full 3D arbitrary acoustic field distributions will open the door to new acoustic devices combining diffraction, scattering and refraction. As a result of advances in metamaterials and the rapidly increasing capabilities for fabricating materials, it is expected that the traditional notion of what constitutes an acoustic device continues to evolve. For instance, in a recent work[37], modern rapid prototyping was used to craft monolithic acoustic holograms, enabling complete control over phase, albeit in a static device. Our quantization approach, along with the functional metamaterial bricks, may enable the future development of fully digital spatial sound modulators, which can be controlled in real-time with minimal resources.

## Methods

**Manufacturing and assembly.** The metamaterial bricks were manufactured from thermoplastics using a 3D printer (ProJet HD 3000 Plus), which has a print resolution of 25 µm. Our bricks are then constituted by two materials[29]: the thermoplastic itself and air. The shoulder bar filets in each brick increased stability during manufacture, as well as contributing to impedance matching. For each position in the grid, we select the phase, among the available ones, which is nearest to the desired analogue value $\varphi(x, y)$. In doing this, we account for the presence of the grid (1 mm wide in Fig. 3c) by making sure that the phase to assign $\varphi_{ij}$ is the one corresponding to an imaginary point at the centre of each brick.

**Numerical simulations.** Full-wave numerical simulations are carried out by the finite element solver in commercial software COMSOL Multiphysics v5.2. Except for the results regarding each brick, all the other simulations are done in 2D. Perfectly matched layers are imposed on the outer boundaries of simulation domains to prevent reflections.

**Transmission measurements.** Transmission amplitude was measured for the bricks by mounting 64 identical bricks in an $8 \times 8$ arrangement. The rest of the grid was filled with absorbing material (to mimic the perfectly matched layers described above). Amplitude was measured with an insertion loss technique using a calibrated B&K microphone (model 4138-A-015) positioned 30 mm above the source array of a 40 kHz transducer (MA40S4S, Murata Electronics, Japan). Results presented in the text refer to bricks #5, #11 and #13.

**Field mapping measurements.** Quantitative measurements were obtained using a calibrated B&K microphone (model 4138-A-015) and a scanning 3D linear stage built in our laboratory. A 2D planar ($8 \times 8$) array of transducers was used to generate a plane wave. Each of the transducers (MA40S4S, Murata Electronics, Japan) has a central frequency of 40 kHz, a beam spread angle of $\pm 40°$and sound pressure levels of $120 \pm 3$ dB (measured on the axis at a $z = 30$ cm). Microphone scans at $z = 10$ mm over the array showed a maximum variation of the phase of 6% across the array, justifying the hypothesis of input plane wave. In our experiments, the transducers are driven at 20 Vpp and operate below the onset of non-linear effects in air. The measured data are normalized with the maximum amplitude of the corresponding simulated data to aid comparison.

**Focusing arrangement.** In this study, we consider the centre of coordinates to be at the centre of the top surface of the higher meta-surface in a stack. To obtain a focus at distance $F_0$ along the axis for an axis-symmetrical arrangement, we used the analogue phase distribution:

$$\varphi(x, y) = \varphi_0 - \frac{2\pi}{\lambda_0}\left(\sqrt{r^2 + F_0^2} - F_0\right) \tag{2}$$

where $r = \sqrt{(x - x_0)^2 + (y - y_0)^2}$ is the distance of the selected point from the axis, located at $(x_0, y_0)$. Since the phase $\varphi(x, y)$ should be modulo-$2\pi$, we selected the central phase $\varphi_0 = \pi$ to alter the phase distribution to the centre of Fig. 3b and reduce the phase changes across the meta-surface. A similar method, which here simplifies manufacturing, has been used with SLMs to increase their operational speed[38]. Note that when using a point source, this phase needs to be complemented with an additional phase distribution, but with the transducer source 4 mm below the meta-surface, this correction was not necessary.

**Data availability.** The authors declare that all data supporting the findings of this study are available within the article and its Supplementary Information files. Further information is available from the corresponding author upon reasonable request.

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

## Acknowledgements

We would like to thank Tom Llewellyn-Jones from the University of Bristol, who helped with the pressure measurements and Luis F. Veloso from Sussex University, who helped with the supplementary video. We acknowledge funding from the EPSRC project EP/N014197/1, 'User Interaction with self-supporting free-form physical objects'.

## Authors contributions

G.M. and S.S. conceived the study and the quantization analysis. Experiments and data analysis were led by G.M., with contributions from all authors. M.C. led the metamaterial simulations with contributions from M.A. and B.W.D. D.R.S. Optimized the manu-facturing technique. G.M. wrote the paper, with contributions from all authors.

## Additional information

**Competing financial interests:** The authors declare no competing financial interests.

**Publisher's note**: 

