## [Peer Review File · Nature Communications]

Reviewers' comments:

Reviewer #1 (Remarks to the Author):

In the revised version of the manuscript, I think the authors have made the point of their work more clear. Although there isn't much novelty in terms of acoustic wave physics, there is an advantage in their methodology which lies in the use of the wavelet transformation-based discretization of the phase profiles. This approach is a good example of application of wavelet transformation in wave engineering and imaging, and is very effective judging from the nice experimental results. Done in ultrasonic frequencies, this work is also closer to meaningful application than preceding works (mostly in kHz). I therefore can recommend its publication on Nature Communications.

Some minor issues:

On page 1, line 37, ref.15 does not describe sub-diffraction focusing.

On page 7, line 243. "Tractor beam" usually refers to a beam of wave generating a force with director opposing the propagation direction of the wave. Here force is in the same direction of the wave, so "tractor beam" is an inaccurate term.

Reviewer #2 (Remarks to the Author):

In the revised version of the manuscript, the authors have clarified some aspects of their work. In particular, it appears that there are two main contributions. 1) Use of wavelet-based compression algorithms to reduce the number of phase steps required in implementation of a graded metasurfaces. 2) Application of the concept of graded metasurfaces to acoustics, including experimental validation.

Regarding the first point, several things are still unclear. For example, comparing their work with Ref. 26, the authors say that in contrast to Ref. 26 which discretizes the number of materials to be used, the present work discretizes phase. However, it is unclear why this is an advantage. In fact, material discretization may be more important from a practical point of view. In general, it is not clear why it is important to design metasurfaces with the minimum possible number of phase steps, considering that, as shown by the authors in Fig. 3, it is quite easy to design constituent elements that cover the entire phase space. It appears to me more meaningful to completely relax the restriction in terms of the number of phase steps and impose only a restriction in terms of the size of the unit cell, which is more important from a practical perspective. Then, the optimum phase profile over the metasurface, which will obviously be different than for subwavelength unit cells, can be calculated through numerical optimization or possibly another smarter approach. In general, the authors make the implicit assumption that image compression algorithms can be efficient for the design of metasurfaces, however it looks that this is not the case, because such algorithms were developed to meet different restrictions and conditions than in the design of metasurfaces.

Regarding the second point, as the authors also mention, there are already several works on gradient acoustic metasurfaces. An experiment in this topic can in principle be interesting for a selective journal, but it is necessary to explain the difference from previous work and the importance for the field of acoustic metasurfaces.

Based on the above remarks, I am afraid I still cannot recommend the paper for publication in its current form.

Reviewer #3 (Remarks to the Author):

The authors address well my raised concerns with clear and convincing evidences. The revised manuscript is indeed more informative in the aspects of the unit cell designs and experimental details to allow reproductive experiments. I think the work is outstanding and deserves the publication in a high-impact journal such as Nature Communications.

Concerning this work's novelty, [REDACTED]
[REDACTED] here's my opinion:

1) Even though various acoustic metasurfaces have been demonstrated to achieve various bending or focusing effects on acoustic beams, the experimental realization of airborne ultrasonic metasurfaces that can achieve advanced functionalities such as object levitation and holographic manipulation is the first time in record for this frequency range.

A very recent publication in Nature on the similar topic (<http://www.nature.com/nature/journal/v537/n7621/full/nature19755.html>) also proves that the topic of this submitted work (realizing spatial sound modulations with complete control over phase) is of major interests and significance. And the submitted work is the first demonstration in the airborne ultrasound range.

The designs (detailed in Supplementary Information S3), while being seemingly similar to the previous space-coiling design, are actually optimized for the much higher frequency range (40 kHz and about 8.6 mm wavelength, an order of magnitude higher than the previous metasurfaces designed for audible sound ranges). None of the designs in the previous works can be straightforwardly applied for this frequency range ([Nat. Commun. 5, 5553 (2014)], [Phys. Rev. Appl. 2, 064002 (2014)] and [Nat. Commun. 7, 11731 (2016)]) given the current limit of additive manufacturing and the minimal feature sizes and material properties.

Besides, as the authors mentioned, the novelty of this submitted work is less on the unit cell designs, but more on the schemes of designing the phase distribution and different representations of the space-phase domain to achieve desired interference patterns in the far field. Similarly, the previously mentioned publication in Nature (<http://www.nature.com/nature/journal/v537/n7621/full/nature19755.html>), while being a nicely demonstration in the higher frequency range, has even less novelty in the aspects of designing phase modulation elements. Therefore, I agree with the authors that the concerns on the novelty are hardly necessary.

2) Efficient and automated discretization schemes are important for designing large scale acoustic metamaterial devices, especially if the dimensions are orders of magnitude of the wavelength. And such schemes can be one of the key steps for realizing truly adaptive spatial sound modulators. The wavelet-based analogue-to-digital conversion method proposed in this work is a clever way for this purpose.

As such, I recommend the publication of this excellent work in Nature Communications.

Reviewer #1 (Remarks to the Author):

In the revised version of the manuscript, I think the authors have made the point of their work more clear. Although there isn't much novelty in terms of acoustic wave physics, there is an advantage in their methodology which lies in the use of the wavelet transformation-based discretization of the phase profiles. This approach is a good example of application of wavelet transformation in wave engineering and imaging, and is very effective judging from the nice experimental results. Done in ultrasonic frequencies, this work is also closer to meaningful application than preceding works (mostly in kHz). I therefore can recommend its publication on Nature Communications.

Some minor issues:

Reviewer's comments	Response
On page 1, line 37, ref.15 does not describe sub-diffraction focusing.	We thank the reviewer for pointing out this misplaced reference. This has now been corrected in the revised manuscript.
On page 7, line 243. "Tractor beam" usually refers to a beam of wave generating a force with director opposing the propagation direction of the wave. Here force is in the same direction of the wave, so "tractor beam" is an inaccurate term.	The particular acoustic field that we use in single-sided levitation consists of a small region of zero (or very low) acoustic pressure, surrounded by a shell where pressure is higher (as shown in Figure 6a). We refer to this as a "bottle-shaped trap" and, because the shell completely surrounds the trapping region, it can be used also with the acoustic radiation force balancing gravity in the direction opposite to the wave propagation. However, since we do not directly present experimental data in this configuration, we have slightly changed the text to "a bottle trap, that can also be used as a tractor beam" (page 7, line 240)

Reviewer #2 (Remarks to the Author):

Reviewer's comments	Response
In the revised version of the manuscript, the authors have clarified some aspects of their work. In particular, it appears that there are two main contributions. 1) Use of wavelet-based compression algorithms to reduce the number of phase steps required in implementation of a graded metasurfaces. 2) Application of the concept of graded metasurfaces to acoustics, including experimental validation.	We would like to add that the concept behind our approach is to reproduce a desired (acoustic) field with a specified precision. The user fixes the precision and our algorithm minimizes the resources (i.e. the types of bricks needed) to achieve the result. This approach leads to optimized brick distributions that are not necessarily uniform in the phase domain (a key difference w.r.t. the literature). The result of this is that there are phases that will be used more than others and this is known before manufacturing. This reduces manufacturing complexity and the idea could be used to discretize any wave array (i.e. equally applicable to optics).
Regarding the first point, several things are still unclear. For example, comparing their work with Ref. 26, the authors say that in contrast to Ref. 26 which discretizes the number of materials to be used, the present work discretizes phase. However, it is unclear why this is an advantage. In fact, material discretization may be more important from a practical point of view. In general, it is not clear why it is important to design metasurfaces with the minimum possible number of phase steps, considering that, as shown by the authors in Fig. 3, it is quite easy to design constituent elements that cover the entire phase space.	The crux here is that the authors in what is now reference [29] did not comment on the efficiency of their implementation. They simply chose a specific approach (i.e. 2 media) and showed that unit cells of different complexity could be manufactured with only two media, in order to cover the required phase range. While revolutionary in optics, this approach was already used in acoustics with some space-coiling metamaterials (which use 3D-printer thermoplastics and air). Our argument is that simply discretizing the constituent parts is necessary but not sufficient to make a system fully digital. In our paper we show that whatever is the type of "bricks" (or unit cells) that are used, our method reduces the resources, simplifies manufacture and makes the process fully digital. We named our work "towards spatial sound modulators" because discretization of phase would be required for these devices. This is true whatever the nature of the bricks. We therefore show an algorithm that could be implemented in control electronics and used with actuated metamaterial bricks (some examples of which we cite at page 9, line 286) to decide which phases the latter should take.

	The text at page 3 (lines 81-83) now links explicitly to Supplementary section S1, which reports further clarification on the difference between our paper and [29]. We have also added a mention to the two materials on page 9, line 309.
It appears to me more meaningful to completely relax the restriction in terms of the number of phase steps and impose only a restriction in terms of the size of the unit cell, which is more important from a practical perspective. Then, the optimum phase profile over the metasurface, which will obviously be different than for subwavelength unit cells, can be calculated through numerical optimization or possibly another smarter approach	While we agree with the reviewer on the practical importance of small unit cells, we still contend that reducing the complexity of metamaterials needs to be achieved both in terms of spatial and phase discretization. Each quantization has its own specific impact on devices and applications. Dealing with each in turn below: We agree that spatial discretization is important and addressed spatial restrictions at page 3, lines 71-79, pushing to the limit the common practice of making the unit cells smaller than half-a-wavelength. We contend that phase discretization is important as:  1) It allows the creation of a fully digital system. 2) It allows the complexity of the metamaterial to be minimized. 3) An optimal compression of the phase pattern at the meta-surface level ensures less information loss in the field. This is confirmed by the results in Supplementary Section S2 and, more in general, by wavefront correlation and information theory. 4) Phase compression naturally forms a step towards what we term a “spatial sound modulator”. This is a device, analogous to a spatial light modulator (SLM) that can manipulate an incident beam to create a vast range of output beams. We have added a clearer statement on the important of phase quantization on page 9, lines 300-302.
In general, the authors make the implicit assumption that image compression algorithms can be efficient for the design of metasurfaces, however it looks that this is not the case, because	There seems to be a misunderstanding here. One of our innovations is the idea of treating a phase map as 2D image, so that we can compress the contained information. It is not at all implicit:

such algorithms were developed to meet different restrictions and conditions than in the design of metasurfaces.	we tested our idea and our experimental results (see figures 6 and S4 and the supplementary video) show that this idea works. We would agree, however, that not all the compression algorithms would work (as discussed in supplemental section S1, for instance, we need an edge-preserving algorithm, like wavelet-based ones). The text at page 3 (lines 84-85) now refers more explicitly to the contribution of the compression process in the AD conversion.
Regarding the second point, as the authors also mention, there are already several works on gradient acoustic metasurfaces. An experiment in this topic can in principle be interesting for a selective journal, but it is necessary to explain the difference from previous work and the importance for the field of acoustic metasurfaces.	Our work differs from the previous ones for many reasons:  1) The previous works show qualitative results, without addressing the issue of precision in reproducing the field, which is crucial in real life applications e.g. audio. We pinpoint how the different design and discretization choices impact on precision. This is clarified at page 4, lines 113-115. 2) We use single metamaterial cells as independent and interchangeable pre-printed “bricks” and not as 3D printed parts of a whole meta-surface. For the first time at 40 kHz. (this is clarified at page 5, lines 150-156) 3) We look thoroughly at gradient acoustic metasurfaces, showing the limitations of generalized Snell’s law (see page 6, lines 195-205). 4) Our quantization algorithm allows us to achieve levitation with meta-materials, again for the first time (see figure 6 and text at page 7, lines 241-243). 5) While we stress that our method could be used with any brick design (see page 4, lines 145-146), and that our design is clearly inspired by the work of others (page 4, lines 136-137), we present 3D-printed bricks with an average (measured) transmission efficiency of 97%. This is not a trivial result, given the current limit of additive manufacturing (page 4, lines 146-150 & page 5, lines 153-154)). 6) We show that stacking different metasurfaces one could achieve a given field

	with less resources, as discussed at page 6, lines 224-226.
--	--

Reviewer #3 (Remarks to the Author):

A very recent publication in Nature on the similar topic (http://www.nature.com/nature/journal/v537/n7621/full/nature19755.html) also proves that the topic of this submitted work (realizing spatial sound modulations with complete control over phase) is of major interests and significance. And the submitted work is the first demonstration in the airborne ultrasound range.	We thank the reviewer for suggesting this reference, that has been added to our list and commented at page 9, line 299.
--	--

REVIEWERS' COMMENTS:

Reviewer #3 (Remarks to the Author):

The authors have addressed my comments satisfyingly.